# Dual Modification of Cassava Starch Using Physical Treatments for Production of Pickering Stabilizers

**DOI:** 10.3390/foods13020327

**Published:** 2024-01-20

**Authors:** Giselle Vallim Correa Ramos, Marya Eduarda Azelico Rabelo, Samantha Cristina de Pinho, Germán Ayala Valencia, Paulo José do Amaral Sobral, Izabel Cristina Freitas Moraes

**Affiliations:** 1Postgraduate Program in Materials Science and Engineering, School of Animal Science and Food Engineering (FZEA), University of São Paulo (USP), Av. Duque de Caxias Norte, 225, Pirassununga 13635-900, SP, Brazil; 2Department of Food Engineering, School of Animal Science and Food Engineering (FZEA), University of São Paulo, Av. Duque de Caxias Norte, 225, Pirassununga 13635-900, SP, Brazilsamantha@usp.br (S.C.d.P.); pjsobral@usp.br (P.J.d.A.S.); 3Department of Chemical and Food Engineering, Federal University of Santa Catarina, Florianópolis 88040-900, SC, Brazil; g.ayala.valencia@ufsc.br; 4Food Research Center (FoRC), University of São Paulo, Rua do Lago, 250, Semi-Industrial Building, Block C, São Paulo 05508-080, SP, Brazil

**Keywords:** ultrasound, antisolvent precipitation, starch nanoparticles, Pickering emulsion

## Abstract

Cassava starch nanoparticles (SNP) were produced using the nanoprecipitation method after modification of starch granules using ultrasound (US) or heat–moisture treatment (HMT). To produce SNP, cassava starches were gelatinized (95 °C/30 min) and precipitated after cooling, using absolute ethanol. SNPs were isolated using centrifugation and lyophilized. The nanoparticles produced from native starch and starches modified using US or HMT, named NSNP, USNP and HSNP, respectively, were characterized in terms of their main physical or functional properties. The SNP showed cluster plate formats, which were smooth for particles produced from native starch (NSNP) and rough for particles from starch modified with US (USNP) or HMT (HSNP), with smaller size ranges presented by HSNP (~63–674 nm) than by USNP (~123–1300 nm) or NSNP (~25–1450 nm). SNP had low surface charge values and a V-type crystalline structure. FTIR and thermal analyses confirmed the reduction of crystallinity. The SNP produced after physical pretreatments (US, HMT) showed an improvement in lipophilicity, with their oil absorption capacity in decreasing order being HSNP > USNP > NSNP, which was confirmed by the significant increase in contact angles from ~68.4° (NSNP) to ~76° (USNP; HSNP). A concentration of SNP higher than 4% may be required to produce stability with 20% oil content. The emulsions produced with HSNP showed stability during the storage (7 days at 20 °C), whereas the emulsions prepared with NSNP exhibited phase separation after preparation. The results suggested that dual physical modifications could be used for the production of starch nanoparticles as stabilizers for Pickering emulsions with stable characteristics.

## 1. Introduction

Starch is a widely utilized carbohydrate source for food and non-food applications and can be found in plants, tubers and roots [1,2]. This polysaccharide is formed from straight chains of amylose (20–30%) and branched chains of amylopectin (70–80%), with the amylose/amylopectin ratio depending on the source [3,4]. Furthermore, starch granules possess a semicrystalline structure that is associated with amylopectin chains’ interactions. These granules are typically classified as A-, B- and C-type polymorphisms, which are typical of cereal, tuber and legume starches, respectively [5].

Native starches have low solubility in cold water, retrograde easily and present low emulsification capacity that limits their food applications [4,6]. An alternative to improve the physicochemical properties of native starches is to modify them through chemical, physical or enzymatic treatments [4,7]. Additionally, there is a strong concern about using safe and environmentally friendly technologies for starch modification. Physical modification of starch is an alternative, since it is a totally green process and free of toxic residues; its products are commonly referred to as clean-label starch. These technologies include modifications with ultrasound (US), heat–moisture treatment (HMT) and nanoprecipitation with antisolvent.

Ultrasound is a nonthermal processing technology that can be used to modify starch through mechanical and cavitation effects. During sonication treatment, the sound waves generate mechanical effects such as agitation and particle dispersion, and the ultrasound energy is transferred through a phenomenon called cavitation. This phenomenon involves the formation and rapid collapse of bubbles in a liquid, resulting in heat and pressure elevations [8]. This technique is useful for modifying the physicochemical and functional properties of starch granules because it offers the advantage of increased selectivity in a short processing time [9].

On the other hand, in HMT, starch is processed at low moisture content (<35%) and high temperatures (T > 90 °C) for a certain period of time (1–16 h). This treatment significantly affects the thermal stability of starch and leads to alterations in structural factors such as crystallinity and granule shape [10,11]. In addition, HMT can increase the linkage of starch chains by disrupting crystalline and helical structures, followed by the reassociation of damaged crystals and increased mobility of the amorphous region that causes the ordering of the double helix [12].

Regarding the antisolvent precipitation method, or nanoprecipitation, this involves gelatinizing starch granules through heating and subsequently precipitation of the biopolymer by adding ethanol dropwise [7]. The antisolvent used in this method is typically a liquid in which the starch is insoluble or has significantly lower solubility. This process is considered safe and nontoxic. Moreover, the nanoparticles produced are amorphous, which increases their flexibility and could aid in the formation and stabilization of the interfacial layer in Pickering-type emulsions [13].

To the best of our knowledge, existing studies have not reported a comprehensive analysis of the microstructural changes and their relationship with the functional properties of cassava starch treated with dual modification (US + nanoprecipitation or HMT + nanoprecipitation). Therefore, this study focused on the correlation between the structural and techno-functional properties of cassava starch modified using two physical methods as a contribution to the application of this nanomaterial as a stabilizer of Pickering emulsions. Thus, we were able to propose a simple and environmentally friendly method for the preparation of tailored cassava starch nanoparticles.

## 2. Materials and Methods

### 2.1. Materials

Cassava starch (Siamar, Neves Paulista, SP, Brazil) was employed as the macromolecule, and canola oil (Liza, Cargil, Mairinque, SP, Brazil), soybean oil (Liza, Cargil, Mairinque, SP, Brazil) and absolute ethanol (99.8%, Êxodo, Sumaré, SP, Brazil) were used in processing. All reagents were of analytical grade.

### 2.2. Physical Pretreatments of Starch

Native cassava starch (12.5 ± 0.1% moisture content (wb); 0.1 ± 0.0% ash; 0.2 ± 0.1% protein content; 0.4 ± 0.1% fiber (db) content) was modified in advance using two separate physical methods: US and HMT.

Native cassava starch was modified with US using the method described by Sujka and Jamroz [14]. A starch suspension (10 wt%) was placed in an ultrasound bath (Q5.9/25A, Eco-Sonics, Indaiatuba, SP, Brazil) at a frequency of 25 kHz and a power of 154 W for 30 min at 25 °C. Afterward, the sample was centrifuged, and the precipitate was dried in an air convection oven (Tecnal, TE-394/3, Piracicaba, SP, Brazil) at 30 °C until it reached a moisture content of ~10% (wb). The dried samples were sieved (100 mesh) [15] and stored in airtight packages.

Starch modification with HMT was conducted following the methodology proposed by Piecyk and Domian [16], with some modifications. The moisture content of the starch was adjusted to 20%, and the samples were stored in airtight packages under refrigeration (~8 °C) for 24 h. After equilibration, the samples were heated at 130 °C for 4 h in an air convection oven, cooled, sieved and stored in airtight packages. The starches were named NCS (native cassava starch), UCS (ultrasonically modified cassava starch) and HCS (HMT-modified cassava starch).

### 2.3. Preparation of Starch Nanoparticles

The production of SNP was carried out according to the methodology described by Ge et al. [17], with some modifications. Cassava starches (native or modified) were dispersed in distilled water (5% *w*/*v*) and gelatinized at 95 °C for 30 min. After cooling, absolute ethanol was added in a 1:1 (*v*/*v*) ratio of water:ethanol, and the mixture was agitated for 12 h at 25 °C. Subsequently, the samples were centrifuged (Eppendorf centrifuge, model 5430R, São Paulo, SP, Brazil) at 3000× *g* for 15 min and washed twice with absolute ethanol. The SNP were freeze-dried in a lyophilizer (LC5500, Terroni, São Carlos, SP, Brazil), crushed, sieved (100 mesh) and stored in hermetically sealed containers. The nanoparticles produced from native starch and starches modified with US or HMT were named NSNP, USNP and HSNP, respectively.

### 2.4. Scanning Electron Microscopy

To analyze the surface of starch samples (native and modified with US or HMT), an electron microscope (SEM HITACHI, TM-3000, Maidenhead, UK) was used. The samples were stored on silica gel and vacuumed for 12 h. A voltage of 15 kV and a magnification of approximately 10,000× were used.

For the observation of SNP obtained from native or modified starches (US or HMT), an SEM FEG microscope (FEI Magellan 400 L, Midland, ON, Canada) with a field emission gun and EDS (energy dispersive X-ray spectroscopy) was utilized. SNP were dispersed in distilled water at a concentration of 0.02% (*w*/*v*), deposited directly on carbon tape along with the stub and dried in a desiccator containing silica gel (25 °C, 12 h). Afterwards, they were covered with gold and subjected to microscopy [18].

### 2.5. Particle Size Analysis and Zeta Potential Measurements

Starch size distribution analysis was performed using a laser diffraction particle size analyzer (Shimadzu, SALD-201V, Kyoto, Japan) and the software Sald Wing, version 1.0 (Sald Wing, Kyoto, Japan). Starch samples were diluted approximately 1000-fold in absolute ethanol. The average size of the particles was expressed by the average diameter (Equation (1)) and calculated from the size distribution curve of the equipment software itself [19].
(1)$$D_{\left[4,3\right]}=\frac{\sum n_{i}d_{i}^{4}}{\sum n_{i}d_{i}^{3}}$$

The average particle size and size distribution of SNP (pH ~7.0) were measured using photon correlation spectroscopy (Malvern, ZetaSizer Ultra, Cambridge, UK), at 25 °C. The samples were diluted in deionized water at a concentration of 0.1% (*w*/*v*) to avoid the phenomenon of multiple scattering of light. Data analyses were performed using software included with the system and are presented as the mean ± standard deviation [20].

The zeta potential of the samples (starches and SNP) (pH ~6.0–7.0) was determined using Zetaplus equipment (Malvern, Zeta Size Ultra, Cambridge, UK) at 25 °C. Samples were diluted in deionized water at a concentration of 0.1% (*w*/*v*) [20]. Data analyses were performed using software included with the system, and the data are presented as the mean ± standard deviation.

### 2.6. X-ray Diffraction (XRD)

Native or modified starches and SNP were stored in desiccators containing silica gel (25 °C) for 10 days. Subsequently, the X-ray spectra of the samples were obtained using an X-ray diffractometer (Rigaku brand, model MiniFlex600, The Woodlands, TX, USA), with a 2θ angle ranging from 4° to 36° and a scan rate of 2°/min at 40 kV and 15 mA [18].

Relative crystallinity (RC, %) was calculated using the ratio between the area of crystalline peaks and the total area under the curve of the XRD spectrum multiplied by 100 [18]. The calculation of the areas was performed using Origin Pro software, version 9 (OriginLab, Sumaré, SP, Brazil).

### 2.7. Fourier Transform Infrared Spectroscopy Analysis (FTIR)

To evaluate the possible changes in the chemical bonds of the starches and SNP, spectroscopy analysis was performed in the infrared region. The samples were kept in a desiccator containing silica gel (25 °C) for 10 days. FTIR spectra were obtained using a spectrophotometer (Perkin-Elmer, Hopkinton, MA, USA) with a UATR (Universal Attenuator Total Reflectance) accessory. Analyses were performed in the spectral range from 400 to 4000 cm^−1^, totaling 16 scans for each sample [18]. FTIR spectra were collected and analyzed with the software Spectrum V 5.3.1 (Perkin-Elmer, Hopkinton, MA, USA). Subsequently, the curves were normalized using Origin Pro software, version 9 (OriginLab, Sumaré, SP, Brazil).

### 2.8. Differential Scanning Calorimeter (DSC)

The thermal properties of starches and SNP were assessed using a differential scanning calorimeter (DSC TA2010, TA Instrument, New Castle, DE, USA). The samples (2 mg) were weighed in an aluminum sample holder and mixed with deionized water at a starch:water ratio of 1:3. The vials were sealed and allowed to be equilibrated at room temperature for 2 h prior to analysis. Scanning was performed at a rate of 10 °C/min, ranging from 30 to 100 °C [21]. An empty aluminum sample holder was used as a reference. Transition temperatures were determined directly from the thermal curves using the Universal Analysis program, version 1.7F (TA Instruments, New Castle, DE, USA).

### 2.9. Solubility in Water (SW) and Swelling Power (SP)

The solubility in water and swelling power were analyzed according to the methodology proposed by Ge et al. [15], with some modifications. Starch and SNP (1 wt%) were placed in a bath for 30 min at a temperature of 90 °C and vortexed every 10 min. Afterwards, the samples were cooled to room temperature and centrifuged at 4000 rpm for 15 min. The supernatant was separated, dried in an oven at 110 °C for 48 h and weighed. The precipitate was also weighed to calculate the swelling power and solubility in water.

### 2.10. Oil Absorption Capacity

To determine the oil absorption capacity, a ratio of 10 mL of soybean oil per 1 g of sample (starch or SNP) was prepared, following the method outlined by Uzomah and Ibe [22]. The mixture was shaken briefly and allowed to stand for 30 min, then centrifuged (3500× *g*, 15 min), and the resulting supernatant was discarded. The residue was weighed, and the excess weight was taken as the oil absorption capacity.

### 2.11. Contact-Angle Measurement

Dispersions containing 4% of gelatinized starch (90 °C, 30 min) or 4% SNP dispersion in water were placed on a glass slide and dried in an oven at 30 °C, until a thin layer of film was formed. A drop of Milli-Q water (5 μL) was placed on top of each film and remained at rest for 45 s. The contact angle was measured using a tensiometer (Theta Tensiometer, Attension, São Paulo, SP, Brazil) [23].

### 2.12. Production of Pickering Emulsions with SNP as Stabilizers

Emulsions were prepared at a ratio of 20:80 (canola oil:water), with 2.4, 3 and 4% (*w*/*w*) SNP as stabilizers (added in the water phase). The dispersed phase (O) was added to the continuous phases (W) at room temperature and homogenized using a rotor–stator homogenizer (Ultra-Turrax IKA, model T25, Labotechnik, Staufen, Germany) at 14,000 rpm for 3 min [24]. Pickering emulsions stabilized with SNP were monitored for 7 days, at 20 °C, away from light, to observe their stability against phase separation.

### 2.13. Statistical Analyses

All analyses were performed in at least triplicate. The results were subjected to analysis of variance (ANOVA), and the means were compared by carrying out the Tukey test (*p* < 0.05) using SAS software, version 9.4 (Statistical Analysis System, São Paulo, SP, Brazil).

## 3. Results and Discussion

### 3.1. Scanning Electron Microscopy

Figure 1 displays SEM micrographs of native and modified starches (Figure 1 a–c), as well as SNP produced from native and modified starches through nanoprecipitation (Figure 1 d–f). NCS presented granules with rounded, oval or truncated oval shapes with very heterogeneous sizes and smooth surfaces without cracks (Figure 1a). Velásquez-Castillo et al. [18] reported similar results for native cassava starch. The starch granules did not show significant differences in shape or on the surface after modification with US (Figure 1b) or HMT (Figure 1c), which indicates a milder change that can be explained by the treatment conditions used in this work. Gunaratne and Hoover [25] reported similar results for ultrasonically treated cassava starch. Dewi et al. [23] modified sago starch using HMT (20% moisture content, 120 °C/1 h) and reported that there was no change in the shape of the granules, but roughness, cracks and cavities were observed on their surface.

SNP showed the existence of laminar aggregates with large particle sizes due to hydrogen bonding interactions, resulting from a high number of hydroxyl groups on the surface of the SNP during nanoprecipitation [17]. After nanoprecipitation, with the formation of aggregate plaques, the interaction between glycosidic chains increases, which decreases the number of hydroxyl groups available for hydration, increasing the hydrophobicity of these particles (SNP) compared to starches [17,26]. In addition, the USNP and HSNP showed greater surface roughness (Figure 1e,f) than NSNP (Figure 1d), which had a smooth plate shape. Ethanol precipitation generally confers surface roughness and expansion of the starch particles, making the structure more porous after nanoprecipitation [27].

The surface roughness of the USNP can be explained by the pre-attack US treatment of the granule, which may have weakened its structure, making it more susceptible to greater damage after nanoprecipitation. As for HSNP, the rough surface can be explained by the evaporation of water during the heat treatment, which induces the molecules of the amylopectin helix double chain to reorganize into a denser packing structure by acting as a barrier to water penetration in the starch granules [23].

### 3.2. Particle Size Analysis and Zeta Potential Measurement

Figure 2 shows the particle size distribution of starches and SNP. NCS showed a bimodal distribution with a lower peak at 2.2 µm and a higher peak at 19 µm (Figure 2a), values similar to those presented by Lima et al. [7] for native cassava starch (2–30 µm). UCS showed a unimodal distribution, with a single peak at 17 µm (Figure 2a), which can be explained by the breakage of agglomerates during ultrasonic processing. HCS showed a polymodal distribution, with peaks at 0.7, 1.9 and 30 µm (Figure 2a); this effect may be due to the weakening of the granules during treatment with heat and moisture.

Similar results have been reported in the literature. Amini et al. [9] sonicated corn starch (25–65 °C, 5–15 min) and observed that US treatment did not considerably change granule size at temperatures below 65 °C. Chandla, Saxena and Singh [28] modified amaranth starch using HMT (28% moisture content, 110 °C for 2.5 h) and reported an increase in mean granule size from 1.4 to 10 µm, which could be attributed to the formation of aggregates or melting during heat treatment.

The SNP also showed different size distributions depending on the treatment to which each starch was subjected. The NSNP exhibited peaks at 38, 207 and 1030 nm (Figure 2b), representing the widest distribution range among the SNP and exhibiting a polymodal distribution. Similar results were found by Lima et al. [7], who reported the size distribution of cassava SNP produced through nanoprecipitation with a slightly acidified ethanol method; the reported sizes ranged from 30 to 100 nm and 200 to 900 nm.

USNP showed peaks at 160 and 950 nm (bimodal) due to the disaggregation effects caused by ultrasound (Figure 2). HSNP exhibited peaks at 117 and 470 nm (bimodal), with sizes smaller than ~700 nm. The formation of smaller particles through HSNP can be explained by the pre-weakening of the granule after HMT modification, which facilitated the breakage and formation of smaller structures during nanoprecipitation.

The polydispersity index (PDI) ranged from 0.80 to 0.42 (Table 1) with a reduction in particles produced after physical pretreatments (US and HMT) and a significant reduction in HSNP (PDI = 0.42). This indicates that the HSNP were particles of more homogeneous sizes than those that were produced using the other tested treatments.

The zeta potential (ZP) of NCS was −50.5 mV (Table 1), a value similar to that reported by Lima et al. [7]. Modified starches also showed high zeta potentials, with no significant difference between them (−44 to −45.3 mV). Conversely, SNP presented much lower ZP values (−2.0 mV on average) (Table 1), which indicates low surface charge, typical of Pickering-type emulsion stabilizers, and the SNP did not differ from each other. Other authors reported different results. Lima et al. [7] reported ZP values for cassava SNP of ~−18 mV. Ge et al. [17] produced SNP from corn, tapioca and sweet potato starches and reported ZP values from ~−14 to −17 mV. These values are higher than those obtained in this work. This difference can be explained by differences in the starch sources or in the SNP production methodology.

### 3.3. X-ray Diffraction (XRD)

All starches exhibited A-type crystal diffraction patterns (Figure 3a), with diffraction peaks at 15°, 17°, 18°, 20° and 23° that were similar to those reported in the literature for NCS [3,7]. The relative crystallinity (RC) of these samples stayed at ~22%. The process parameters applied in the pre-modification of starch did not alter starch crystallinity compared to that of the native starch. Similar results have been reported by Rahaman et al. [3].

On the other hand, other authors reported a slight reduction in RC for US-treated starches [3,9], suggesting that the US may have damaged the crystalline structure of the starch granules, which did not occur in this work. Ali et al. [6] reported more drastic changes for HMT-treated lotus seed starch, with an increase in RC from 37.1 to 44.2%, and explained this fact as being due to the displacement of the double helices towards the interior of the crystallites during the HMT, leading to a more compact and ordered arrangement, a behavior also not observed in this work.

SNP showed a V-type crystal structure with a small diffraction peak at 13° and a halo at 20° (Figure 3b). This crystalline structure is typical of starches with low crystallinity [7]. The SNP presented much lower RC than the starches, staying around 3%. This reduction in the RC value was related to the breakdown of the starch crystalline structure after gelatinization and precipitation with antisolvent. Similar results were reported by Lima et al. [7] (RC = 1.0–2.5%). Furthermore, the strong peak intensity of V-type and the low relative crystallinity of the SNP suggest that gelatinization was complete, and there was amylose/amylopectin dispersion of the starch granules after nanoprecipitation. In addition, the V-type crystal structure was formed through the rapid formation of amylose–ethanol complexes [27].

### 3.4. Fourier Transform Infrared (FTIR) Spectroscopy Analysis

The FTIR spectra of the starches, as well as the SNP, are depicted in Figure 4. The absorbance spectra were quite similar for all samples. This observation is likely attributable to the absence of new chemical groups introduced into the starch molecules through the modifications induced by the US, HMT or nanoprecipitation.

The peak at 3440 cm^−1^ corresponds to the stretching vibration of the −OH groups, related to intra- and intermolecular hydrogen bonds. The bands between 2800 and 3000 cm^−1^ are attributable to CH stretching vibrations, which reflect the hydrophobicity and lipophilicity of the starch [29]. The band at ~1640 cm^−1^ is associated with the moisture content of the samples and relates to the vibrations of water molecules adsorbed in the noncrystalline region [30]. The peaks at 1010, 1080 and 1150 cm^−1^ are attributable to the C–O and C–C stretching vibrations of the polysaccharide molecules [31]. Furthermore, the absorption bands between 950 and 1150 cm^−1^ are related to the vibrations of the glycosidic bonds C–O, C–O–C and C–O–H [32].

After starch pre-modification (US or HMT), there was an increase in peak intensity at 2928 cm^−1^, which reflects an increase in the lipophilicity of the modified starch. Similar results were reported by Wang et al. [33] for ultrasonically treated potato starch. Furthermore, after nanoprecipitation, the peak at 1659 cm^−1^ showed a reduction, suggesting that fewer water molecules hydrated the SNP, i.e., an indication that the SNP has higher hydrophobicity in water when compared to starches. Lima et al. [7] reported similar results for cassava and potato SNP; furthermore, the bands at 1022 and 1047 cm^−1^ are indicative of amorphous and crystalline structures, respectively. All SNP showed an increase in the 1018 cm^−1^ band when compared to starches, indicating an increase in the amorphous content of this material, which corroborates the XRD results.

### 3.5. Differential Scanning Calorimetry (DSC)

The differential scanning calorimetry (DSC) curves of NCS (Figure 5a) showed a peak (T_p_) temperature of 69.3 °C (Table 2), with no significant difference for UCS. These values are similar to those reported by Lima et al. [7] for native cassava starch. HCS, on the other hand, showed significantly lower values of T_p_ (67.8 °C), which can be explained by the weakening of the granules after heat treatment.

All SNP showed DSC curves related to amorphous materials, with no gelatinization transition (Figure 5b). These results are in agreement with the XRD, confirming that the nanoprecipitation of the starches promoted the breaking of the crystalline structure.

Similar results were reported by Lima et al. [7], who produced cassava and potato SNP through nanoprecipitation and did not observe peaks in the DSC curves of these samples, which corresponded to amorphous materials.

### 3.6. Solubility in Water, Swelling Power and Oil Absorption Capacity

The solubility in water (SW) and swelling power (SP) values at 90 °C of the starches and SNP are shown in Figure 6a,b. The SW of NCS was ~84%, a result that was slightly higher than that reported by Agyemang et al. [34] (SW ~74%). Differences in these values may be due to the particle size, amylose/amylopectin ratio, gelatinization temperature, different periods of plant maturation or different types of cultivars or methods of extraction.

After US pretreatment, the samples did not show differences in SW values when compared to native starch. Different results were reported by Sujka and Jamroz [14], who modified potato, wheat and rice starches using US and reported an increase in SW in all cases, indicating that the increase was due to damage to the crystalline molecular structure of starch and to the binding of water molecules to the free hydroxyl groups of amylose and amylopectin through hydrogen bonds. In our work, the crystallinity remained the same after US (Figure 3b), which corroborates the SW value of UCS.

HMT pretreatment significantly decreased the SW of both starch (84–71%) and SNP (78–60%). During the HMT process, the mobility of molecules increases due to the interaction of amylose–amylose and amylose–amylopectin chains, which decreases the number of hydroxyl groups available for hydration and diffusion of amylose–amylopectin molecules [23]. These results corroborate with Dewi et al. [23], who modified sago starch using HMT at 20% moisture content and 120 °C/1 h and reported a reduction in SW from 38.5 to 25% after modification.

Overall, the SW values of starches (84, 83 and 71%) were always higher than those of their SNP counterparts (78, 79 and 60%), which could indicate that starch is more soluble in water. This result agrees with the FTIR results, which indicated that SNP have more lipophilic characteristics than starches. The smooth and highly hydrophilic surface of starch can promote its increased solubility in water.

The swelling power (SP) of the NCS was ~16 g/g, which corroborates the value of ~17 g/g that was reported by Dudu et al. [35]. HMT pretreatment increased the SP of both starch (16–21 g/g) and SNP (21–24 g/g) compared to their native counterparts. This may occur due to amylose–amylopectin chain rearrangement during HMT, allowing greater starch swelling.

US pretreatment increased the SP of cassava starch due to the depolymerization of the chains during this physical treatment, which would allow greater swelling of the granules. In general, SP values showed an increase after nanoprecipitation, from 16 to 21 g/g for NSNP and 21 to 24 g/g for HSNP, with the exception of USNP, which did not show a significant difference from UCS (average value of ~18 g/g).

Regarding the oil absorption capacity (OAC) of NCS, it was similar to that reported by Agnes et al. [36] (Figure 6c). Modification with HMT increased the ability of starch to absorb oil (169%) compared to NCS or UCS (~150%). This indicates that the HCS became much more lipophilic. This behavior may be associated with granular expansion and the release of helical structures from the starch granules after treatment. These results agree with those presented in the FTIR spectra (Figure 4), where there was an increase in the intensity of the band related to the lipophilic of cassava starch after physical modifications.

The nanoprecipitation method increased the OAC values of all samples when compared to their starch counterparts, from 147 to 193% for native, 154 to 205% for US and 167 to 255% for HMT, indicating the greater lipophilicity of SNP. This behavior can be explained by the formation of aggregates in the form of plates after starch nanoprecipitation, which increases the lipophilicity of the particles. In addition, ethanol precipitation imparts surface roughness and expansion to the starch particles, further strengthening internal capillary strength and oil affinity [26].

Furthermore, the formation of the V-type structure allows the amylose helix, with a hydrophilic outer surface and an inner hydrophobic helical cavity [37], to assemble into a crystalline structure and further accommodate hydrophobic compounds, such as oil molecules, mainly driven by hydrophobic interactions, hydrogen bonding and diffusion concentration [27]. Similar results were reported by Feng et al. [27], who produced corn starch SNP through nanoprecipitation and reported a 75 to 160% increase in OAC after nanoprecipitation. These results corroborate the results of the FTIR, which indicated the increased hydrophilicity of the SNP in relation to starches.

### 3.7. Water Contact Angle Measurement

Measuring the water contact angle enables a qualitative estimation of changes in the hydrophobicity of starches and SNP (Figure 7). Among all analyzed samples, NCS had the smallest contact angle (26.3°). Pretreatment with US and HMT led to an increase in the contact angle of the starch (~38 and 49°, Figure 7), suggesting enhanced hydrophilicity. This finding is in line with the results obtained from OAC and FTIR.

After nanoprecipitation, the SNP presented drastically higher values (~68 to 77°) than the starches (~26 to 49°), and the HSNP presented the highest contact angles. This behavior can be explained by the V-type polymorphism exhibited by SNP, where hydroxyl glycosyl groups are located on the outer surface of the helix of V-amylose complexes, while the inner surface is coated with methylene groups and glycosidic oxygen, resulting in a more hydrophobic cavity similar to that of cyclodextrins [37]. Furthermore, the surface roughness of the modified SNP (Figure 1e,f) increased their lipophilic affinity, allowing the formation of a more resistant barrier to water droplets on the surface through the development of aggregate plaques after nanoprecipitation. These results are in agreement with the OAC analysis (Figure 6c).

Furthermore, for all samples, the water contact angle remained relatively stable with time (0 to 45 s) (Figure 7). Similar results were found in the literature. Ge et al. [17] produced SNP from tapioca, sweet potato and corn starches using the nanoprecipitation method and reported contact angles of ~84 to 94°. Dewi et al. [23] modified sago starch using HMT and OSA and reported higher contact angles after double modification, indicating that HMT helped to make the starch more hydrophobic. In addition, these authors observed that for the native starch, the angle decreased during the observation period of the analysis from 0 to 90 s, which did not occur as visibly for the modified starches.

Additionally, measurement of the water contact angle allows assessment of the interfacial wettability (water–oil) of the particles. For Pickering-type emulsions, this is a fundamental factor for understanding the behavior of the particles as a stabilizer, as any hydrophobic variation of the particles would affect their wetting properties [37]. In general, SNP with an almost neutral contact angle (~90°) have a greater potential to stabilize this type of emulsion [17].

### 3.8. SNP as Pickering Emulsion Stabilizers

The SNP produced were applied as a stabilizer of oil-in-water Pickering emulsions in various concentrations (2.4–4 g SNP/100 g emulsion) (Figure 8). The emulsions were monitored for phase separation on days 0 and 7 after production.

Oil–water emulsions with NSNP or USNP exhibited phase separation at all tested concentrations; however, the emulsion prepared with NSNP showed more pronounced separation after preparation. USNP promoted stability for up to 24 h at the highest concentration (4%). The emulsion stabilized with HSNP showed stability at higher concentrations (4%), and phase separation did not occur over the storage period. Higher concentrations of SNP allow more efficient coating at the droplet interface [17]. These results support the data obtained from the water contact angle measurements, where HSNP had the highest values and exhibited the most effective stabilization. In addition, the smaller particle size of HSNP and the surface roughness contribute to more hydrophobic behavior for these SNP, further enhancing their stabilization properties. In fact, surface roughness plays a crucial role in particle adsorption, movement and interactions at fluid interfaces, promoting greater particle adhesion at the interface [38]. In other words, surface roughness strongly fixes the particle contact lines. Therefore, the combined properties of the particles modified with HMT + nanoprecipitation allowed the formation of a more stable interfacial layer, with the particles more firmly adsorbed, providing greater stability to the system.

Ge et al. [17] also produced SNP from corn, tapioca and sweet potato starches and applied these particles in Pickering-type emulsions with different concentrations of SNP (0.25–2 wt%) and oil fractions (0.25–0.75); however, these authors reported phase separation of the emulsion as early as the first day of monitoring for all SNP produced. This difference may be due to the different sources of starch used for SNP production, the water–oil ratio used in the formulation and, most likely, the fact that this study included no pretreatment of the starches for the production of SNP. This hypothesis is strong because, in our study, the native cassava SNP also failed to efficiently stabilize the emulsion, i.e., the effect of physical pretreatment (HMT) clearly improved this property of the SNP.

## 4. Conclusions

By changing the properties of starch nanoparticles through surface modifications, particle size control and the production of effective changes in the lipophilicity of the material, it was feasible to produce stabilizers for o/w Pickering emulsions. All SNP produced were amorphous, and the dual modification promoted increases in roughness and lipophilicity and a significant reduction in average sizes. In addition, HSNP stood out because they had the smallest size and largest OAC. These properties yielded the formation of a robust and stable interfacial layer that prevented droplet coalescence and provided emulsion stability. The US + nanoprecipitation combination was not able to stabilize the Pickering emulsion, whereas the HMT + nanoprecipitation combination produced nanoparticles with synergistic properties that stabilized the emulsion for 7 days at 20 °C. In addition, this biomaterial is classified as a clean-label product and safe for the consumer.

## Figures and Tables

**Figure 1 foods-13-00327-f001:**
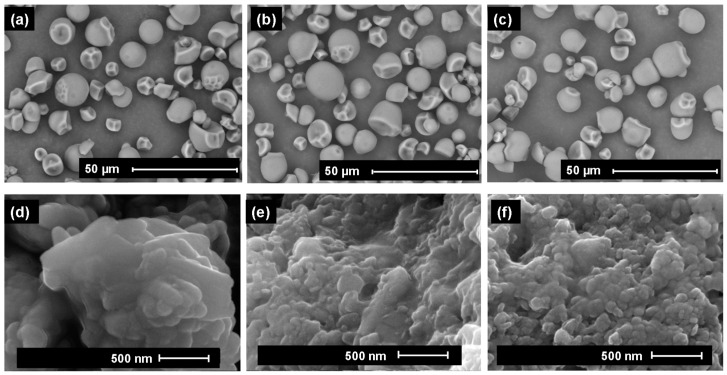
SEM analysis of (**a**) native cassava starch (NCS) and cassava starch modified with (**b**) US (UCS) or (**c**) HMT (HCS) and starch nanoparticles produced from native starch (**d**) (NSNP) and starch modified with (**e**) US (USNP) or (**f**) HMT (HSNP). Scale bar: 50 µm or 500 nm.

**Figure 2 foods-13-00327-f002:**
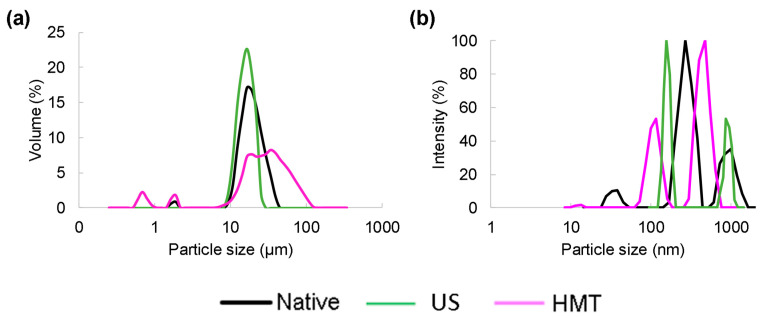
(**a**) Size distribution of native cassava starch (NCS) and cassava starch modified with US (UCS) or HMT (HCS) and (**b**) starch nanoparticles produced from native starch (NSNP) and starch modified with US (USNP) or HMT (HSNP).

**Figure 3 foods-13-00327-f003:**
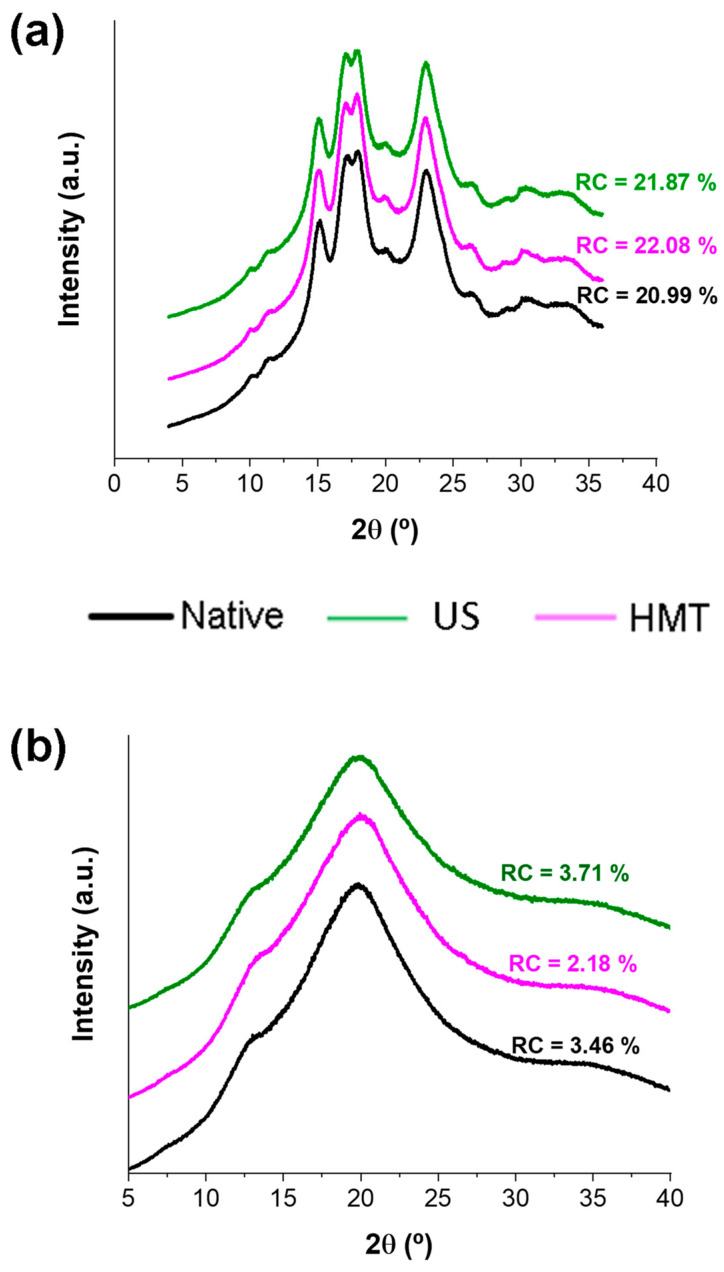
(**a**) XRD and relative crystallinity of native cassava starch (NCS) and cassava starch modified with US (UCS) or HMT (HCS) and (**b**) starch nanoparticles produced from native starch (NSNP) and starch modified with US (USNP) or HMT (HSNP).

**Figure 4 foods-13-00327-f004:**
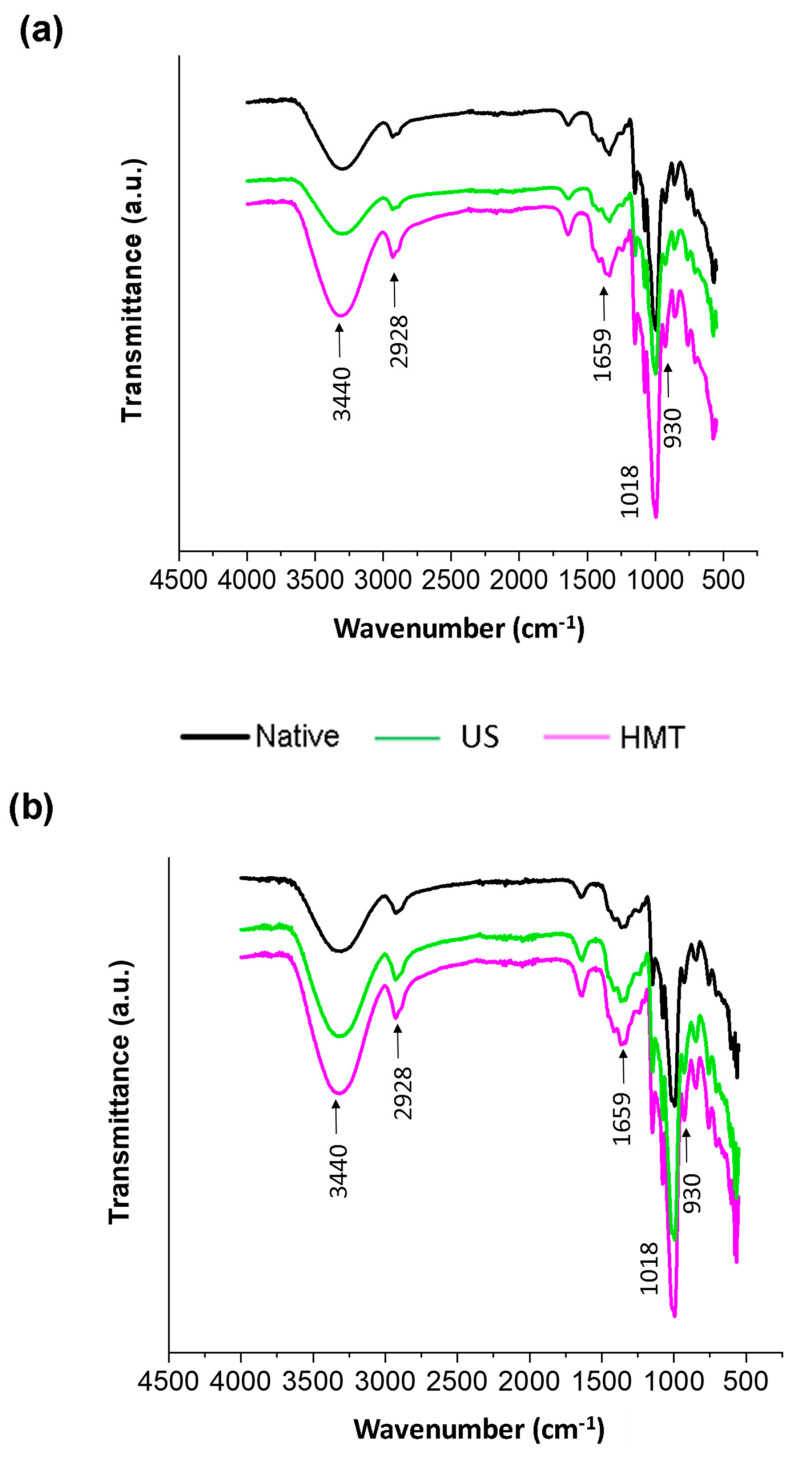
(**a**) FTIR spectra of native cassava starch (NCS) and cassava starch modified with US (UCS) or HMT (HCS) and (**b**) starch nanoparticles produced from native starch (NSNP) and starch modified with US (USNP) or HMT (HSNP).

**Figure 5 foods-13-00327-f005:**
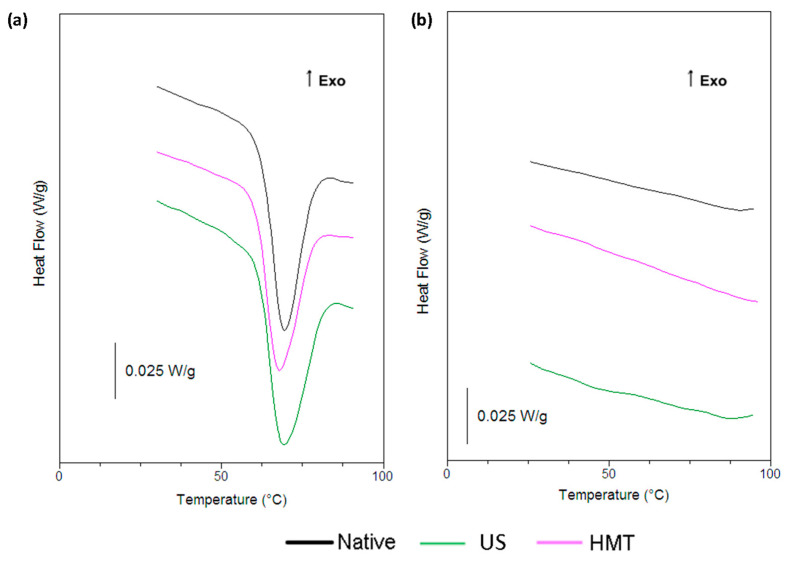
(**a**) Heat-flow curves of native cassava starch (NCS) and cassava starch modified with US (UCS) or HMT (HCS) and (**b**) starch nanoparticles produced from native starch (NSNP) and starch modified with US (USNP) or HMT (HSNP).

**Figure 6 foods-13-00327-f006:**
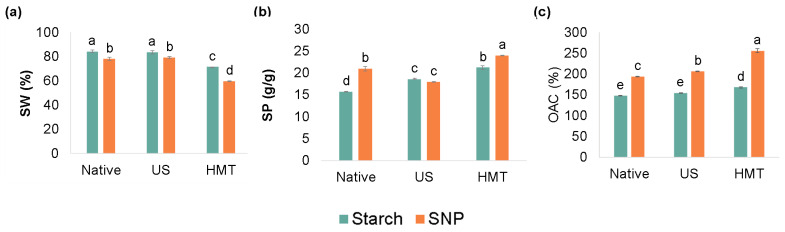
Solubility in water (**a**); swelling power (**b**) and oil absorption capacity (**c**) of native (NCS), US-modified (UCS) and HMT-modified (HCS) cassava starches—blue; starch nanoparticles produced from native (NSNP), US-modified (USNP) and HMT-modified (HSNP) starches—orange. Different lowercase letters indicate a significant difference (*p* < 0.05).

**Figure 7 foods-13-00327-f007:**
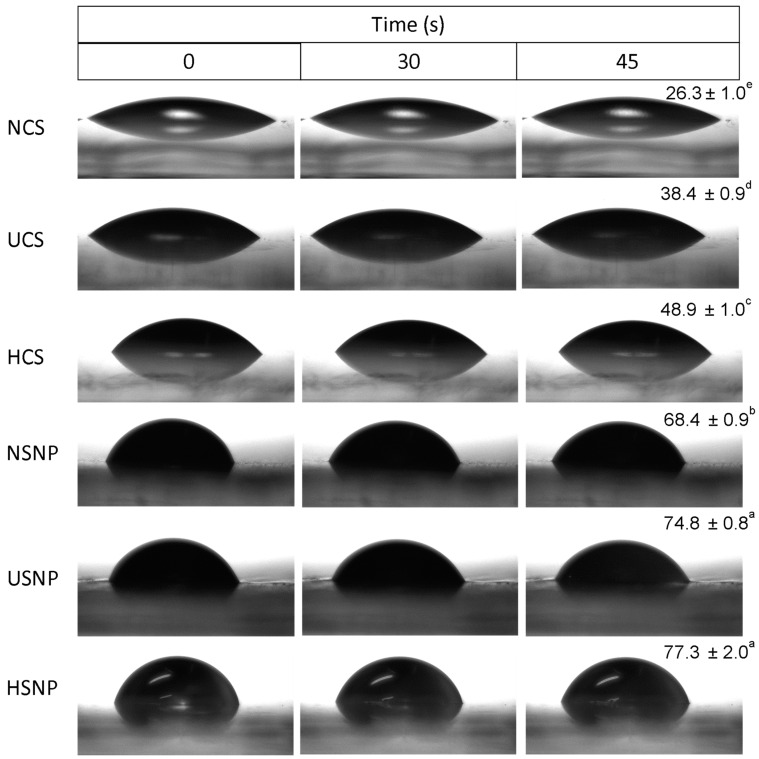
Images of contact angles of native cassava starch (NCS) and cassava starch modified with US (UCS) or HMT (HCS) and starch nanoparticles produced from native starch (NSNP) and starch modified with US (USNP) or HMT (HSNP), as a function of time. Different lowercase letters indicate a significant difference (*p* < 0.05).

**Figure 8 foods-13-00327-f008:**
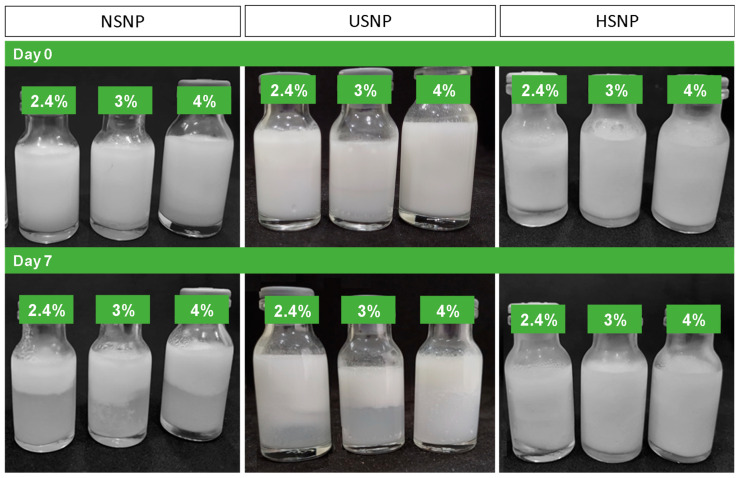
Pictures of O/W emulsions (20% canola oil) stabilized with NSNP, USNP and HSNP, respectively, at concentrations of 2.4%, 3% and 4% (*w*/*w*), in a freshly emulsified state and after 7 days of storage.

**Table 1 foods-13-00327-t001:** Polydispersity index (PDI) and zeta potential (ZP) of starches and starch nanoparticles.

Sample	PDI	Zeta Potential (mV)
NCS	-	−50.5 ± 1.3 ^c^
UCS	-	−44.0 ± 1.0 ^b^
HCS	-	−45.3 ± 2.9 ^b^
NSNP	0.80 ± 0.11 ^a^	−1.0 ± 0.2 ^a^
USNP	0.65 ± 0.12 ^a,b^	−1.7 ± 0.5 ^a^
HSNP	0.42 ± 0.01 ^b^	−3.4 ±0.6 ^a^

NCS: native cassava starch; UCS: ultrasonically modified starch; HCS: HMT-modified starch; NSNP: native starch nanoparticle; USNP: ultrasonically modified starch nanoparticle; HSNP: HMT-modified starch nanoparticle; -: not determined. Different lowercase letters in the same column indicate a significant difference (*p* < 0.05).

**Table 2 foods-13-00327-t002:** Thermal parameters of native cassava starch (NCS) and cassava starches modified with ultrasound (UCS) or HMT (HCS).

Sample	T_0_ (°C)	T_p_ (°C)	T_c_ (°C)	∆H (J/g)
NCS	62.8 ± 0.2 ^a^	69.3 ± 0.1 ^a^	83.5 ± 0.9 ^a^	14.4 ± 0.8 ^a^
UCS	62.2 ± 0.3 ^a^	69.3 ± 0.1 ^a^	84.8 ± 1.0 ^a^	13.9 ± 0.3 ^a^
HCS	61.0 ± 0.1 ^b^	67.8 ± 0.2 ^b^	83.1 ± 0.9 ^a^	13.6 ± 0.1 ^a^

T_0_ initial gelatinization temperature; T_p_: peak gelatinization temperature; T_c_: conclusion gelatinization temperature of peak; ΔH: gelatinization enthalpy. Different lowercase letters in the same column indicate a significant difference (*p* < 0.05).

## Data Availability

Data is contained within the article.

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
