# Peer review of "Dual Modification of Cassava Starch Using Physical Treatments for Production of Pickering Stabilizers"

_foods, 2024, doi:10.3390/foods13020327_

Round 1
Reviewer 1 Report
Comments and Suggestions for Authors
In this paper, the physical modification methods of cassava starch were carried out. The content of the study is detailed, but the summary and condensation in the abstract are not in place.
This study lists the research results and does not dig deeply into the above results because the structure of cassava starch has undergone important changes, which is also the significance of physical modification. The improvement of the amorphous region of cassava starch by physical modification is the main reason for the improvement of the stability of Pickering emulsion.
1. In Lines 90-92, how about the US temperature? It is necessary to control the temperature in the study, otherwise, is it caused by the modification of starch by ultrasonic or by high-temperature gelatinization? How to compare with heat moisture treatment?
2. Line 94, Why did the author choose a 100-mesh sieve? Whether it is too large? This is relevant to the particle size analysis described later. In this process, starch will agglomerate. How about the native cassava starch?
3. Line 98, how about the refrigeration temperature? Whether the starch appear aging or the phenomenon of regeneration?
4. Line 167, There is no need for the temperature to rise to 130 ºC.
5. Line 107, :ethanol, enter a space.
6. Line 117, vacuumed.
7. Line 118, was used.
8. Line 132, were carried out.
9. Line 144, was obtained.
Comments on the Quality of English LanguageGood.
Reviewer 2 Report
Comments and Suggestions for Authors
The manuscript presents extensive description of the experimental investigation about specific modifications of cassava starch materials aimed to be applied as Pickering emulsion stabilizers. The paper is well-focused, properly detailed and contains important research outcomes. This investigation might be of considerable interest to the readers of MDPI Foods, and be useful in view of further industrial application prospects.
Before accepting the manuscript however, some issues should be clarified:
1. The conclusion “These properties yielded the formation of a robust and stable interfacial layer that prevented droplet coalescence and provided emulsion stability”(lines 485-486) should be explained and substantiated in more details in the Results and Discussion section.
2. A list of the used abbreviations will be helpful, e.g. SW, SP, SNP, US, HMT, OAC, etc.
3. Some misprints and English-language irregularities should be corrected, e.g. line 40 “…retrogradation…”; lines 185-186 “…a thin layer of film formed.”; line 188 “…and the software of the equipment itself…”; line 312 ”…molecules due to through the...”
Comments on the Quality of English Language
Some misprints and English-language irregularities should be corrected (see item 3 in the review).
